# The Effects of Neurodevelopmental Treatment-Based Trunk Control Exercise on Gross Motor Function and Trunk Control in Children with Developmental Disabilities

**DOI:** 10.3390/healthcare11101446

**Published:** 2023-05-16

**Authors:** Miho Park, Jeongseon Kim, Changseon Yu, Hyoungwon Lim

**Affiliations:** 1Department of Physical Therapy, Graduate School, Dankook University, Cheonan 31116, Republic of Korea; 2Department of Physical Therapy, Gangdong University, Eumseong-gun 27600, Republic of Korea; 3Department of Rehabilitation Medicine, Dongguk University Ilsan Hospital, Goyang 10326, Republic of Korea; nerve@dumc.or.kr; 4Dankook University Disabled Rehabilitation Research Institute, Department of Physical Therapy, Dankook University, Cheonan 31116, Republic of Korea

**Keywords:** pediatric rehabilitation, trunk control, Gross Motor Function Measure, Segmental Assessment of Trunk Control

## Abstract

Background: Good trunk control is essential for higher developmental stages as the trunk is activated first when movement occurs, providing stability for the head and extremities. Purpose: To determine if neurodevelopmental treatment-based trunk control exercise (NDT-TCE) is effective in improving gross motor function and trunk control in children with developmental disabilities (DD). Materials and Methods: Twenty children with developmental disabilities were randomly assigned to the NDT-TCE (12 children) and control (8 children) groups. Results: After the intervention; the NDT-TCE group showed improvement in GMFM (Gross Motor Function Measure; except for the GMFM-E dimension) and SATCo scores. The control group showed improvement in GMFM-A; B; C; and total scores; as well as static and active control of SATCo. The NDT-TCE group had a significant improvement in the GMFM B dimension and total score compared to the control group. The NDT-TCE group showed a significant improvement in static and active control of SATCo compared to the control group, but there was no significant difference in reactive control. Conclusions: The NDT-TCE intervention specifically improved GMFM-B and trunk control scores. Therefore, NDT-TCE can be applied as a trunk-focused intervention for children with DD who have difficulty controlling their trunk.

## 1. Introduction

Trunk control is crucial for children’s development, as it provides stability for the movement of the head and extremity [1]. Better trunk control enables a higher developmental stage to be reached. Good trunk control enables stable head and hand movements, allowing the child to grip a toy more rapidly and hold it more stably. Stable head movements create stable eye movements, which increases visual and cognitive development, enabling a wider range of play activities [2]. When trunk control near the pelvis is improved, movement of the upper and lower extremities is diversified [3]. However, unlike typically developing children, children with developmental disabilities (DD) are slow, inefficient, and produce unstable movements [4]. Children with DD have poor stability and difficulty controlling the trunk, in that when they try to reach for toys, trunk activation does not occur first, rather the arms move first [5]. Unstable movements limit children’s movement and play and lead to delays in various areas including motor skills, learning skills, cognition, and activities of daily living [6]. This results in restrictions in children’s daily lives and affects their future school life [7].

Trunk control exercises (TCE) can improve motor skills in children with DD [1,8]. TCE can improve gross motor functions (sitting, standing, walking, etc.), static and active balance, and trunk muscle strength and can reduce the risk of falls [9]. TCE in children with DD improves their delayed motor function and alters the developmental curve in a positive direction [10]. TCE use various methods such as functional electrical stimulation, gross motor task training, hippo-therapy, neurodevelopmental treatment (NDT), progressive resistance exercise, balance training, treadmill training, trunk-targeted training, virtual reality, and visual biofeedback [9]. Overall, NDT is the most popular intervention used in pediatric physiotherapy [9,11]. NDT focuses on appropriate postural control and selective movement using the Bobath concept [12]. NDT and Bobath are used synonymously [13]. The Bobath concept focuses on the facilitation of muscle activation for proximal control (trunk, pelvis, and head) and leads to appropriate postural control [14].

Looking at previous studies that applied NDT to children with cerebral palsy, as a result of NDT for trunk control, the Pediatric Balance Scale, 1-Minute Walking Test, Timed Up and Go test, and muscle strength all reported significant improvements compared to the control group [15,16]. NDT interventions for CP for 12 weeks showed significant functional improvements in all dimensions but no significant improvement in Gross motor function measure (GMFM) dimension E (walking, running, and jumping) [17]. However, in a review of 17 randomized studies in which NDT was applied, 6 studies reported uncertain effects, while 6 of the remaining 11 studies reported conflicting results, indicating uncertainty about the effectiveness of NDT [11]. On the other hand, Lee et al., [18] found that NDT is effective in children having DD with or without CP. However, they did not focus on TCE, only confirming the GMFM total score, and did not confirm the effectiveness in trunk control of DD [18]. See also Arndt et al. [19] reported an improvement in motor function when an NDT-based trunk activation protocol was applied to infants with motor impairment. They did not divide subjects into CP and DD (excluding CP), only the GMFM total score.

Children with DD and CP exhibit different types of impairments in trunk control. CP limits body movement due to clinical characteristics such as spasticity, making it difficult to maintain a sitting position and to control the trunk [20]. In contrast, DD is associated with low trunk muscle tone [21] and a lack of stability in the trunk, resulting in more trunk sway compared with typically developing children and difficulty in controlling the trunk control and maintaining a sitting position [22]. Trunk control is therefore a very important factor in DD. TCE for DD can improve motor function [1] and increase the developmental stage [10]. However, there is a lack of studies on NDT-based trunk control exercises (NDT-TCE) for DD. Therefore, the purpose of this study was to determine whether the NDT-TCE, which was studied for its effectiveness in CP, is also effective in improving trunk control in children with DD.

## 2. Materials and Methods

### 2.1. Participants

The study included 28 children with DD and poor trunk control who received physical therapy services in the community, and children who were found to be unable to walk or unable to walk independently (at Daejeon, Sejong, and Cheonan city in Republic of Korea). Inclusion criteria were as follows: age ≤ 8 years, diagnosis of developmental delay or developmental disability due to delayed one or more motor milestones by a pediatrician or rehabilitation physician, at Gross Motor Function Classification System (GMFCS) level ≥ II. Exclusion criteria were as follows: a diagnosis of CP at GMFCS level I or independent walking, musculoskeletal deformations that can affect posture control, and having undergone orthopedic surgery within the last six months.

Of the 23 patients enrolled, 5 were excluded due to various reasons, including 3 who could walk independently, 1 diagnosed with CP, and 1 with musculoskeletal deformities. Random allocation resulted in 12 patients in the experimental group (NDT-TCE group) and 11 patients in the control group receiving traditional physical therapy. However, 3 patients withdrew from the control group, resulting in a final sample of 8 patients (Figure 1). The general characteristics of the participants are presented in Table 1.

The study was approved by the Dankook University Review Board and performed in accordance with the Helsinki Declaration (Approval No. 2020-12-015-001). We obtained written informed consent from the children’s parents. The study was conducted over four months from February to May, 2021.

### 2.2. Procedure

Participants were recruited by posting notices in community physical therapy rooms and randomly divided into experimental and control groups. Pre- and post-assessments were performed using the GMFM and Segmental Assessment of Trunk Control (SATCo) tests, and the order of measurements was randomized. The evaluation was performed by a pediatric physical therapist who had completed training on the evaluation tool. The experimental group received NDT-TCE, and the control group received conventional physical therapy only. The intervention was conducted for six weeks, twice a week for 30 min in both groups.

Since cognitive aspects were not assessed in this study, to address this issue, the evaluation was performed as follows. During the GMFM assessment, children who had difficulty following the therapist’s instructions were encouraged to move using a favorite toy or treat. If they were still unable to complete the task, the therapist instructed them to do the task and then checked if they could do the task independently. For items that could not be rated directly, we requested videos from parents and rated them ourselves. In contrast, the SATCo test required participants to reach for a toy, which motivated all participants.

### 2.3. Intervention

#### 2.3.1. Neurodevelopmental Treatment-Based Trunk Control Exercise (NDT-TCE)

As an intervention method in this study, the handling and facilitation principle of NDT was used to induce trunk uprightness, weight bearing, and elongation [19]. The starting position was sitting posture with support to the maximum trunk control area that could be adjusted by the patient, according to SATCo measurements. For example, children who could not adjust their static trunk control below the ribs were supported below the ribs, and children who did not need support started in an independent sitting position.

As part of the intervention method in this study, we used the handling and facilitation principle of NDT to induce trunk uprightness, weight bearing, and elongation. The NDT intervention was developed in three stages. In the first stage, the children were induced to play with toys in front of them to facilitate dynamic co-activation of the flexor and extensor trunk muscles and allow the trunk to remain upright without tilting sideways in the sagittal plane. In the second stage, the children were allowed to play with toys located at their side to induce weight shift while maintaining dynamic co-activation of the flexor and extensor trunk muscles and facilitate elongation on the weight-bearing side. In the third stage, maintenance of the elongation of the weight-bearing side was induced by placing the toy at the side of the elongation in a position of approximately 45 degrees, inducing the child’s trunk rotation in the horizontal plane and At this time, rotation of the trunk was facilitated on the weight-bearing on the side of elongation. We checked the child’s movements and trunk alignment during the intervention, ensuring that the eyes were horizontal, the head was not tilted, the trunk muscles were active, and the weight-bearing side had trunk elongation. The intervention was conducted in a stepwise manner, with each stage building on the previous stage. We analyzed the data collected during all stages of the intervention to evaluate its effectiveness. The intervention stages are shown in Figure 2.

#### 2.3.2. Conventional Physiotherapy

Conventional physiotherapy is a common treatment performed in the pediatric physiotherapy room. The conventional physiotherapy performed in this study included upper and lower extremity stretching, strength strengthening, balance training, active and passive ranges of motion exercises, and gross motor function such as sitting, standing, and walking.

### 2.4. Measurements

#### 2.4.1. Gross Motor Function Measure-88 (GMFM-88)

GMFM is an assessment tool designed to measure gross motor function in children [23,24]. It is an accurate and reliable tool to measure changes in gross motor function after any intervention targeted to improve gross motor functions. [25]. The 88 items of the GMFM are measured by observation and scored according to a 4-point scale. The scoring criteria were as follows: 0, did not attempt; 1, tried (activity at 10%); 2, tried but not perfect (activity at >10%); and 3, perfect (activity at 100%). The 88 items were grouped into five dimensions: (A) lying and rolling (17 items), (B) sitting (20 items), (C) crawling and kneeling (14 items), (D) standing (13 items), and (E) walking, running, and jumping (24 items) [23]. The GMFM dimensions score is a percentage score. Each score is the child’s score/maximum score × 100%. The total score is obtained after summing the scores of each dimension and dividing by five. Total number of five GMFM dimensions (A, B, C, D, and E) [23]. When the GMFM score is lower, the skill level is lower.

#### 2.4.2. Segmental Assessment of Trunk Control (SATCo)

SATCo is an evaluation tool used to assess segmental control of the trunk in children who cannot sit independently or sit with an impaired posture/trunk. The overall reliability score is 0.80 or higher. SATCo is a reliable and validated test that assesses static, dynamic, and reactive control of trunk in a segmental fashion instead of testing it as single uni. The trunk is segmented into the head, upper thoracic, middle thoracic, lower thoracic, upper lumbar, lower lumbar and full trunk. In each trunk segment, the manual support areas are the shoulders, axillae, inferior scapula, lower ribs, below the ribs, and pelvis, or no support is provided and pelvic/thigh straps are removed [8]. The measurement method is that the child sits on a bench, the foot supported on the floor or a stable surface, and the pelvis and thighs are held in a neutral position with straps connected to the bench. The test is performed in a cranial to caudal direction. Manual support is provided by the evaluator, horizontally, around each trunk segment. There is no additional help other than the evaluator’s manual support [26]. Static control is maintained using a neutral head up or a neutral vertical trunk for 5 s. Active control is to maintain a neutral vertical position while turning the head 45 degrees or stretching the arms from side to side. Reactive control was to maintain or quickly return to a neutral vertical sitting position when lightly pressing down on the trunk segment with the fingertips [26]. The child sits upright with a neutral pelvis position maintained using straps around the waist and attached to the bench (Figure 3).

At the level of each trunk segment, the presence (✓) or absence (-) of control, or not tested (NT) status, are recorded and scored. Each trunk segment with a presence (✓) is scored for statistical analysis [26]. Scores assigned to each segment are as follows: 1, head control; 2, upper thoracic control; 3, mid-thoracic control; 4, lower thoracic control; 5, upper lumbar control; 6, lower lumbar control; 7, insufficient full trunk control; and 8, complete trunk control. Score assignments for scores 7 and 8 are divided according to whether a child performs perfectly when performing a sitting position independently or not. Score 7 indicates that the child cannot fully control the full trunk independently without hand support. Score 8 indicates a child can fully control the full trunk independently without any help [8]. The SATCo was recorded on video, and the evaluator reviewed this video to determine the SATCo score [25]. This procedure is the same as that used in previous studies [8,25,26,27].

### 2.5. Statistical Analysis

The general characteristics of the study subjects were analyzed using descriptive statistics. The normality test using the Shapiro-Wilk test did not indicate normal distribution of the data. The Wilcoxon Signed-Rank Test was used to determine the difference in GMFM and SATCo scores before and after the intervention, both in the NDT-TCE experimental group and the control group. To compare the changes in GMFM scores between the two groups before and after the intervention, ANCOVA was conducted with all values of age and baseline as covariates to exclude the difference in age between groups and the effect of baseline values on post-intervention values in repeated measurement data. All analyses were performed us-ing SPSS version 26 (IBM Corp, Chicago, IL, USA). Statistical significance was set at *p* < 0.05.

## 3. Results

### 3.1. General Characteristics of the Participants

Table 1 shows the general characteristics and diagnoses of the study subjects. The descriptive characteristics of the subjects can be found in Table 1. The experimental group consisted of six males and six females, and the control group consisted of five males and three females. The gender distribution between the groups was homogeneous. The average and standard deviation of age was 21.92 ± 12.27 months in the experimental group and 32.75 ± 21.95 months in the control group, indicating that the age of the control group was higher and that the distribution of age between groups was not homogeneous.

### 3.2. Comparison of GMFM before and after Intervention within Groups

In the NDT-TCE group, the GMFM score increased in A, B, C, and D dimension after the intervention, and was statistically significant (*p* < 0.05). The GMFM E dimension increased to 2.02 after the intervention, but was not statistically significant (*p* > 0.05). The total GMFM score also increased significantly after intervention (*p* < 0.05). On the other hand, in the control group, only GMFM A, B, C dimension and total score significantly increased after intervention (*p* < 0.05). In the GMFM comparison be-tween groups, the NDT-TCE group significantly increased the GMFM B dimension and total score than the control group (*p* < 0.05) (Table 2).

### 3.3. Comparison of Changes in SATCo Scores within and between Groups before and after Intervention (Scores: %)

SATCo score was significantly increased in all three conditions in the NDT-TCE group (*p* < 0.05). In the control group, there was a significant increase in the remaining two conditions except for the reactive control. In the comparison of SATCo scores between groups, the NDT-TCE group increased significantly in the static and active control than the control group, and there was no significant difference in the reactive control (Table 3).

## 4. Discussion

The experimental group that underwent six weeks of NDT-TCE training improved in all dimensions of GMFM. Statistically significant improvements were observed in all dimensions except for the E dimension. Compared with the control group, the experimental group improved and showed significant improvements in the GMFM B dimension, total GMFM score, static control, and active control. This is consistent with the results of many previous studies that reported improved GMFM scores after NDT intervention [16,17,18,24,28,29,30].

For CP, NDT-based posture and balance training improved both the GMFM score and the alignment of posture [16]. When NDT with focused on trunk control is provided for CP, GMFM-88, postural assessment scale (PAS), pediatric balance scale (PBS), and trunk impairment scale (TIS) scores are improved [28]. The findings of this study align with earlier research that demonstrate the effectiveness of NDT-TCE in improving trunk control and gross motor function in children with CP and DD. Intensive NDT has been shown to significantly improve GMFM score in all DD and CP [18]. Children with hypotonicity have shown improved GMFM scores and trunk control after dynamic weight-bearing exercise using the NDT principle [30]. NDT is therefore effective for children with DD, CP, and hypotonicity, which supports the hypothesis of this study that NDT is effective for CP and will also be effective for DD. In children with DD, the trunk muscle activation is delayed or impaired as a result of low trunk muscle tone, which leads to unstable movements [4]. For DD, where muscle activity is low due to weak muscle strength and low muscle tone, better strength and tension improve stability and gross motor function [1]. NDT has been shown to improve trunk strength [16], and in this study, children with DD who received NDT-TCE improved their trunk control potentially due to increase in muscles strength, tone, and appropriate co-activation of flexor and extensors. NDT-TCE, which improves trunk strength and induces activation of trunk muscles, was shown to improve motor function and stability, and gross motor function. Previous studies have suggested that NDT-based trunk-focused interventions for CP improve GMFM scores, the trunk control-related assessment tools TIS and TCM scores, and alignment of posture [15,16,28]. In this study, using SATCo, a trunk control evaluation tool, we found that all areas of SATCo improved after NDT-TCE. In addition, static control and active control were significantly improved. The SATCo is an evaluation tool that examines trunk control in a sitting position on a bench [26], and scores improve as control of trunk segments proximal to the pelvis is improved [31]. In addition, the better the trunk control near the pelvis, the better the alignment of trunk control-related assessment tools [32]. As static balance and dynamic balance improve, TIS and TCM are improved [33]. This is consistent with the improvement in SATCo and trunk control-related test scores in previous studies, which shows that NDT-TCE was effective for trunk control, as in previous studies.

Ahmed M et al. [11] used facilitation as one of the NDT methods to conduct core stability exercises focusing on dynamic activities of the trunk. As a result, the activation of trunk agonists and antagonists improved sitting posture and body control in the NDT group. Activation of these trunk agonists and antagonists leads to trunk joint activation and lumbar stability [34]. Additionally, trunk dynamic co-activation of trunk flexors and extensors has been shown to improve trunk movement [35]. In this study, we speculate that improved trunk control may be attributed to appropriate co-activation of trunk muscles. Dynamic sitting postures improved sitting balance, further improving SATCo and total GMFM scores in the experimental group, and GMFM B showed particularly significant improvements. This supports our hypothesis that NDT-TCE is more effective in improving the static and active control scores of GMFM B and SATCo compared to typical physical therapy intervention. In this study, it is considered that there was no significant difference in GMFM D and E dimensions since GMFCS level I children were excluded.

In this study, NDT-TCE, a treatment focused on trunk control, improved the SATCo and GMFM scores. Improvements in SATCo scores and GMFM scores are also correlated with DD, especially in GMFM B. This is consistent with previous studies exhibiting a high correlation between SATCo scores and total motor function and GMFM B [26]. The 11-month developmental stage of starting to stand or walk independently is associated with reactive control of the SATCo, and the eight-month developmental stage, which is sitting independently or sitting in a chair, is associated with the static and active control of the SATCo. A previous study found a moderate to good correlation between static and active segmental trunk control status and gross motor skills in a sitting position at eight months of age in typically developing infants [8]. The subjects of the present study were children who started to sit or were able to sit at the developmental stage of eight months. Therefore, we believe that static and active control were more significantly improved in the experimental group at the developmental stage of the subjects in this study.

Children’s developmental stages and ages are also highly correlated with SATCo scores, and GMFM score is also highly correlated with motor developmental stage and age [36]. In this study, there was a significant difference in age between groups despite randomization. Therefore, after adjusting for age, the results before and after the intervention were compared. As a result, the NDT-TCE group significantly increased the GMFM B dimension and total score than the control group. In a prior study, children aged 18–23 months and 30–35 months scored the same in GMFM A and B dimensions, but at 30–35 months they scored higher in the C, D, and E dimensions [37]. This is consistent with the higher pre-intervention scores of the control group who were older in this study. However, in this study, the experimental group that underwent NDT-TCE had an average age of 21.92 months and had more score changes than the control group with an average of 32.75 months. This suggests that NDT-TCE produces a faster change in scores, suggesting that NDT-TCE is an effective intervention. In the GMFM B dimension, the experimental group scored lower before the intervention than the control group, but the experimental group scored higher after the intervention. This supports that NDT-TCE was effective in the GMFM B dimension. The amount of %-GMFM change that translates into everyday functioning will depend on the individual and the specific activity in question [38]. In general, a change of 5% or more in the GMFM score is considered clinically meaningful and can result in improvements in everyday activities such as sitting, standing, walking, and running. However, it is important to consider other factors such as the individual’s baseline function, their overall health and medical status, and their goals for treatment [38]. Therefore, in this study, the NDT-TCE group showed a lower age than the control group, but even after adjusting for age, an improvement of more than 5% was observed in the GMFM B dimension compared to the control group, which could be clinically meaningful.

The limitation of this study is that it was difficult to test normality due to the small number of samples, so it is difficult to generalize the effects of NDT focusing on trunk control regulation in children with DD. In addition, the intervention was short-term, and the evaluator was not blinded. Further studies will require larger sample sizes, in which the data are normally distributed, in order to be able to generalize the effects of the intervention. Furthermore, the intervention periods should be longer and the evaluators should be double-blinded.

## 5. Conclusions

This study conducted NDT-TCE intervention for six weeks in children with DD who were unable to walk independently. GMFM (except GMFM E) and SATCo scores improved in the experimental group, post-intervention. GMFM A, B, C, GMFM total, and static and active control of SATCo improved in the control group, post-intervention. There was a significant difference between GMFM B, total GMFM, and static and active control of SATCo scores in the experimental group. SATCo scores were highly associated with improved trunk control and GMFM dimension B scores, and the higher the SATCo score, the higher the GMFM score. Trunk-focused interventions provide significant improvements in trunk control and have been confirmed to improve gross motor function by stabilizing the sitting posture. Therefore, we propose NDT-TCE as a treatment for children with DD who have difficulty controlling their trunk in clinical settings.

## Figures and Tables

**Figure 1 healthcare-11-01446-f001:**
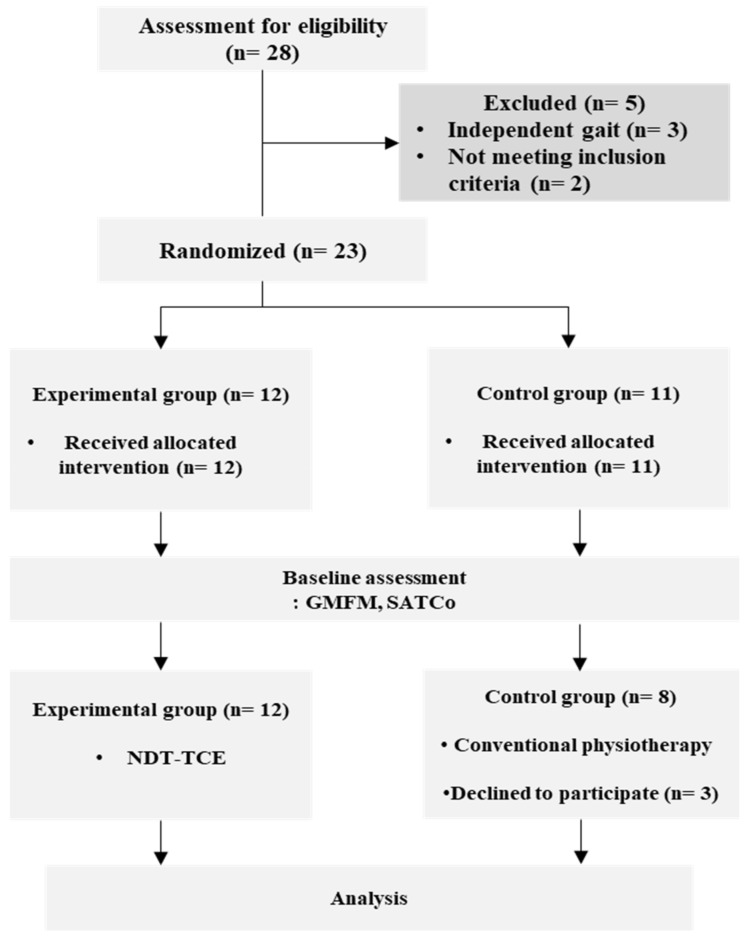
Flowchart of the study.

**Figure 2 healthcare-11-01446-f002:**
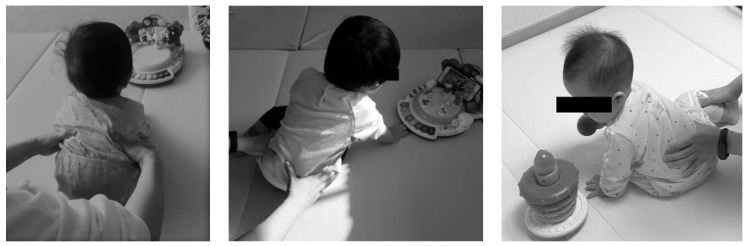
Three stages of NDT-TCE in children with developmental disabilities. Starting from the left, the first, second, and third stages.

**Figure 3 healthcare-11-01446-f003:**
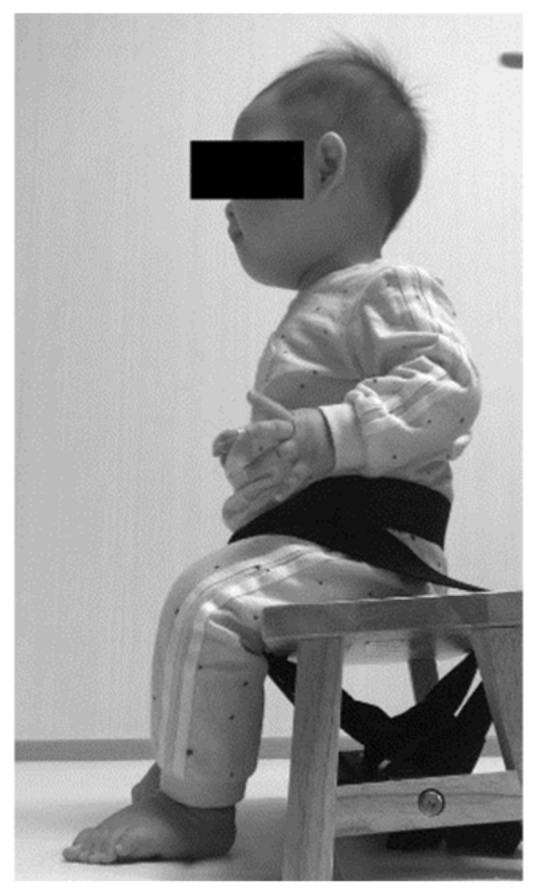
A child undergoing the SATCo.

**Table 1 healthcare-11-01446-t001:** General characteristics of participants (N = 20).

General Characteristics	Experimental Group	Control Group	*p*-Value
N (Male/Female)	12 (6/6)	8 (5/3)	0.927
Mean age (months) (±SD)	21.92 ± 12.27	32.75 ± 21.95	0.046
Diagnosis			
Unspecified (developmental disorder)		3	1
Tetralogy of Fallot		2	-
Brain damage		1	1
Hypotonicity		2	-
Lennox-Gastaut syndrome		1	-
Coffin-Siris syndrome		-	1
Down syndrome		-	2
Chromosomal abnormality		1	1
Suspected chromosomal abnormalities		2	-
Cytomegaloviral disease		-	1
Congenital toxoplasmosis		-	1
Mean GMFCS level (±SD)		3.75 ± 0.97	3.38 ± 1.30
I		-	-
II		1	3
III		4	1
IV		4	2
V		3	2

**Table 2 healthcare-11-01446-t002:** Comparison of changes in GMFM scores within and between groups before and after intervention (Scores: %).

		Mean ± SD			
		Pre-Intervention	Post-Intervention	z	*p*-Value ^(1)^
GMFM A	Experimental group	73.04 ± 26.00	87.73 ± 14.53	−2.668	0.008
	Control group	79.71 ± 29.22	85.29 ± 22.67	−1.841	0.006
	*p*-value ^(2)^		0.065		
GMFM B	Experimental group	41.67 ± 27.93	67.36 ± 27.99	−3.064	0.002
	Control group	54.36 ± 32.16	61.89 ± 34.03	−2.524	0.012
	*p*-value ^(2)^		0.002		
GMFM C	Experimental group	15.48 ± 28.18	33.73 ± 36.92	−2.366	0.018
	Control group	40.19 ± 34.50	47.66 ± 39.44	−2.371	0.018
	*p*-value ^(2)^		0.423		
GMFM D	Experimental group	5.98 ± 14.43	16.45 ± 26.25	−2.207	0.027
	Control group	16.35 ± 21.58	22.45 ± 24.32	−1.826	0.068
	*p*-value ^(2)^		0.861		
GMFM E	Experimental group	0.00 ± 0.00	2.02 ± 5.46	−1.342	0.180
	Control group	3.99 ± 6.85	6.95 ± 9.06	−1.826	0.068
	*p*-value ^(2)^		0.893		
GMFM Total	Experimental group	27.23 ± 16.82	40.96 ± 20.22	−3.061	0.002
	Control group	38.92 ± 22.46	44.84 ± 24.01	−2.521	0.012
	*p*-value ^(2)^		0.046		

*p*-value ^(1)^: Wilcoxon signed rank test pre and post intervention within group. *p*-value ^(2)^: Repeated measured ANCOVA of between groups. Covariate: months, baseline.

**Table 3 healthcare-11-01446-t003:** Comparison of changes in SATCo scores within and between groups before and after intervention (Scores: %).

		Mean ± SD		z	*p*-Value ^(1)^
		Pre-Intervention	Post-Intervention		
Static control	Experimental group	4.92 ± 2.11	7.00 ± 1.13	−3.097	0.002
	Control group	5.63 ± 2.26	6.38 ± 2.00	−2.121	0.034
	*p*-value ^(2)^		0.004		
Active control	Experimental group	3.17 ± 2.76	6.33 ± 2.15	−3.078	0.002
	Control group	4.88 ± 2.48	5.63 ± 2.77	−2.121	0.034
	*p*-value ^(2)^		0.005		
Reactive control	Experimental group	0.83 ± 1.27	3.42 ± 3.29	−2.384	0.017
	Control group	0.88 ± 1.25	2.00 ± 2.27	−1.841	0.066
	*p*-value ^(2)^		0.239		

*p*-value ^(1)^: Wilcoxon signed rank test pre and post intervention within group. *p*-value ^(2)^: Repeated measured ANCOVA of between groups. Covariate: months, baseline.

## Data Availability

Data will be available from the corresponding author and will be released on reasonable request.

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
