# Peer review of "The Effects of Neurodevelopmental Treatment-Based Trunk Control Exercise on Gross Motor Function and Trunk Control in Children with Developmental Disabilities"

_healthcare, 2023, doi:10.3390/healthcare11101446_

Round 1
Reviewer 1 Report (New Reviewer)
First of all, I have to say that I found this manuscript interesting enough to be considered for publication in this journal. However, before doing so, the authors should take into account a series of considerations:
- The document must be uploaded in doc format as it is in pdf format and with change control enabled.
- The words “Developmental disabilities” and “Neurodevelopmental treatment” should not be considered as keywords as they are included in the title of the article. Authors should look for other suitable keywords that are not mentioned in the title.
- It would be advisable to include the practical implications of the study as well as future lines of research. The latter, taking into account the small size of our population, would need to be extrapolated to larger population sizes.
Nevertheless, my sincere congratulations.
Author Response
Thank you very much for your careful review of our paper.
Keywords have been changed.
sincerely,
Reviewer 2 Report (New Reviewer)
Hello,
I have attached manuscript document with my edit (look for comments/edits).
I suggest few minor edits to manuscript. I have highlighted and provided my comments to each section (track changes). Please respond, modify, and or address these concerns. This will improve the reading and quality of this work.

I moderate edits to manuscript to improve clarify and quality.
The content is good, but need some work, specially when discussing previous literature and long sentences.
Author Response
Thank you very much for your careful review of our paper.
SG7: None of the children received Botox injections in our study.
SG16: We did not include a reference in our study.
In our study, the participating children described the treatment they were receiving in the hospital.
Sincerely,
This manuscript is a resubmission of an earlier submission. The following is a list of the peer review reports and author responses from that submission.
Round 1
Reviewer 1 Report
Park et al. presented an interesting study on NDT-TCE for DD children. The manuscript was well-written, and the methods and results were clearly presented and discussed.
- The authors should explain the signifiant difference in the scores of GMFM and SATCo between experimental and control groups in pre-intervention, as they are supposed to be grouped randomly.
- BACKGROUND in Abstract should be discussed a little bit more to hightlight the significance of the study.
- The abbreviation should be explained on the first occurrence, such as GMFM, SATCo, CP.
- The variance of participant ages is too high, particularly in the control group, and the authors should disscuss it more.
- The table2 between 268-281 should be reformatted.
- The double-line (second line) in table4 should be reorganized to make it consistent.
- The authors may disscuss more about the improvements of GMFM, SATCo score in the control group.
Author Response
We sincerely thank the reviewers for carefully reviewing the paper despite its shortcomings. The answer is attached as a file.
Sincerely,
Prof. Hyoungwon Lim.

Reviewer 2 Report
Major comments:
Page 3, line 93 ff: the inclusion and exclusion criteria are not clear to me. Was the developmental delay really only defined on the motor level without any respect to the cognitive level? How was “delay” in milestones defined quantitatively (for example in % GMFM)? Children with GMFCS level can walk unaided at normal speed and only look for support at stairs or higher burden, why were they included? Exclusion criteria: “a diagnosis of CP at GMCFS level I or independent walking”: that means that more severe CP cases were included?
Page 3, line 105: what was the method of randomization leading to such asymmetric result as 12:8?
Page 4, line 129: it must be clearly stated here that the evaluator was not blinded (and not only at the end under discussion). This is a major problem for the validity of the findings! Why was the evaluator not blinded?
Page 4, line 145: “Our analyses begun in the third stage”: I do not understand this sentence! I understand that the experimental treatment developed in 3 stages, but which “analyses”? And why only in the third stage? How many therapists were involved in the experimental treatment? How were they trained?
Page 5, 171ff: How many therapists were involved? This is a broad mixture of methods with very variable effectiveness: were they kept constant? Were they comparable in all children and therapists?
Page 6, line 232 ff: was a trained biostastician involved? Did you consider correction for multiple testing?
Page 7, table 1: the experimental group is nearly 1 year younger than the control group, is this difference statistically significant? The normal increase in the GMFM values with age is not linear, but it is steeper in younger ages and then flattens later: what does this mean for the interpretation of your data?
Further table 1: the list of diagnoses in the present form is not really helpful, in part this is a list of symptoms or histories, in part of genetic entities. Also, the variability of disabilities is very broad (more than 50% in GMCFS IV and V). I am sure that many of these patients also have significant cognitive problems, and this has to be considered as an important determinant of understanding and motivation of the children which is important for the treatment results (and must be included in the analysis).
Page 8, table: what is the clinical significance of statistically significant %-GMFM changes? How large must a %-GMFM change be to translate into everyday functioning?
Page 10, tables: In Methods you described that you performed ANCOVA to control for age and other factors, however, you mentioned its results only shortly at the end of discussion. It is necessary to present the full findings of the ANCOVA here under results. You also must declare, if the preconditions for an analysis of variance are fulfilled (for example homogeneity of variances between groups).
At the end of the discussion, you mention the major weaknesses of this study yourself. It is my advice to stress the pilot character or your study even more. For me it is very questionable if on the basis of only 20 (12 and 8) patients with very different underlying diseases and necessarily also cognitive abilities such detailed conclusion concerning the GMFM subdomains are possible and allowed, at all.
Page 14 ff, List of references: The format of your references is not consistent. Please correct all references according to the requirements of the journal.
Formal and minor comments:
Page 2, line 58ff: what is the sense of these additional references in text form?
Page 2, line 71-73: in the first part of the sentence “uncertain effects” clearly means contradictive or insignificant effects. But what does “conflicting results” in the second part mean? How can conflicting results indicate effectiveness of NDT?
Page 2, line 78: does “DD (excluding CP)” mean “DD without signs of CP”?
Page 3, line 93: “children who found it difficult to walk independently”. This wording sounds strange to me: apparently the children could not walk independently, but the finding was one of the investigators, so better “children who were found to have problems with/or could not walk independently”
Page 3, line 105: the findings are in table 1, not 2.
Author Response

(The authors gave the same response as above.)

Round 2
Reviewer 2 Report
major comment:
1) I understand from the response of the authors that initially patients were equally randomised in the treatment and control groups, but that later during the study more dropped out in the control than the treatment group giving rise to the asymmetric numbers of 12 and 8 patients. But this is not what is shown in fig 1 and described in the text: there the asymmetric randomisation is reported after the drop out! It must be made clear what has happened: 28 have been randomised (14 and 14) followed by a asymmetric drop out? And this must also be described under methods and in the abstract.
2) it must be clearly indicated under methods, results and in the abstract that the evaluator and the treating physician were the same person and not blinded. This is necessary for the readers to clearly understand the most important weakness of this study!
minor comment:
Separation signs (-) must be checked in the abstract and other parts of the reviewed text.